# Rituximab, Intravitreal Bevacizumab and Laser Photocoagulation for Treatment of Macrophage Activation Syndrome and Retinal Vasculitis in Lupus: A Case Report

**DOI:** 10.3390/ijms24032594

**Published:** 2023-01-30

**Authors:** Marina Ikić Matijašević, Paula Kilić, Lucija Ikić, Iva Galić, Vlatka Brzović Šarić, Edvard Galić

**Affiliations:** 1Internal Medicine Clinic, University Hospital Sveti Duh, 10 000 Zagreb, Croatia; 2Department of Internal Medicine, School of Medicine, University of Zagreb, 10 000 Zagreb, Croatia; 3Department of Anatomy and Physiology, University of Applied Health Sciences, 10 000 Zagreb, Croatia; 4University Eye Clinic, University Hospital Sveti Duh, 10 000 Zagreb, Croatia

**Keywords:** systemic lupus erythematosus, retinal vasculitis, macrophage activation syndrome, hemophagocytic lymphohistiocytosis, rituximab, bevacizumab, anti-VEGF, laser photocoagulation

## Abstract

Systemic lupus erythematosus (SLE) most commonly manifests as mild to moderate disease with severe manifestations such as diffuse alveolar hemorrhage, central nervous system vasculitis, macrophage activation syndrome (MAS) or retinal vasculitis (RV) with visual disturbances occurring in a significantly smaller proportion of patients, most of whom have a poor outcome. Macrophage activation syndrome and RV are insufficiently early and rarely recognized presentations of lupus—consequently there are still no treatment recommendations. Here we present the course of diagnosis and treatment of a patient with an SLE flare that resulted in both life-threatening disease (MAS) and vision-threatening disease (RV). The patient was successfully treated with systemic immunosuppressives, a high dose of glucocorticoids and rituximab (RTX), in parallel with intraocular therapy, intravitreal bevacizumab (BEV) and laser photocoagulation.

## 1. Introduction

Systemic lupus erythematosus is a complex systemic autoimmune connective tissue disease with a chronic relapsing–remitting course and many different presentations and complications, with a spectrum of disease ranging from mild to life threatening. It can affect any organ, including part of the eye. The prevalence of eye involvement in SLE according to Turk et al. is 31% [1]. Keratoconjunctivitis sicca (secondary Sjögren’s syndrome) is the most common ocular manifestation of SLE and lupus retinopathy in patients with active SLE, and it is one of the most common vision-threatening complications, with an incidence between 15.8% and 29% [2,3,4]. Severe retinopathy can cause visual loss in up to 80% of cases, with a decrease in final visual acuity in half of the patients [5]. High SLE disease activity can also result in a potentially fatal syndrome known as MAS or secondary hemophagocytic lymphohistiocytosis (HLH), which has a prevalence in SLE ranging from 0.9% to 4.6%. However, it is important to note that MAS is an underdiagnosed condition despite its high mortality rate [6,7]. The syndrome itself is characterized by persistent fever, cytopenias, hepatosplenomegaly, lymphadenopathy, elevated liver transaminases, ferritin, triglycerides, d-dimers, and decreased albumin and fibrinogen. High mortality in MAS is due to the hyperinflammatory condition itself that leads to organ dysfunction, which may progress to multi-organ failure. It is also a consequence of the fact that MAS is rare and the diagnosis itself is complicated and based on multiple criteria which are not yet validated in adult SLE population, namely the HLH-2004 clinical criteria, the European League Against Rheumatism (EULAR)/American College of Rheumatology (ACR)/Pediatric Rheumatology International Trials Organization (PRINTO) in 2016, HScore (a score for the diagnosis of reactive hemophagocytic syndrome), and MAS complicating juvenile SLE criteria [8,9,10,11,12]. Herein, we present the case of MAS and RV in a patient with an SLE flare. After concomitant treatment with a laser photocoagulation, an intravitreal monoclonal antibody against vascular endothelial growth factor (anti-VEGF) BEV and an anti-CD20 monoclonal antibody RTX, the patient is in SLE remission with maintained visual acuity.

## 2. Case Description

A patient who has been suffering from SLE for the past 10 years and who has stopped taking hydroxychloroquine and a small dose of prednisone for the last 6 months was admitted to our clinic due to their first SLE flare, which involved 6 weeks of a fever > 38.5 °C, an acute cutaneous lupus rash, oral ulcers, leukopenia, normocytic anemia, thrombocytopenia and polyarthralgia (Table 1). Upon admission, MAS was suspected. The patient also had ultrasound-proven hepatomegaly and splenomegaly, cervical and axillary lymphadenopathy, significantly increased liver enzymes, a headache without visual disturbances, decreased albumin and fibrinogen, and increased ferritin, triglyceride and d-dimers (Table 1). The calculated HScore upon admission was 223 points with a 96–98% probability of HLH.

A further extensive diagnostic workup was undertaken. Obtained blood and urine cultures were negative, and viral panels for Herpes Simplex, Herpes Zoster, EBV, cytomegalovirus, Hepatitis B and C and HIV viruses were negative. An MSCT of the thorax, abdomen and brain (including a cerebral angiography) did not show evidence of inflammation or other pathologies; only hepatosplenomegaly and axillary lymphadenopathy were found. Immunophenotypic analysis of cells isolated from bone marrow and a bone biopsy excluded lymphoma and leukemia but also did not show hemophagocytic cells. Unfortunately, in our country, we are unable to determine soluble CD25 and CD163 or NK cell activity. Furthermore, subsequent immunological laboratory findings were strongly positive for ANA, with high anti-dsDNA, very low levels of complement (C)3 and C4, and positive antiphospholipid antibodies (Table 1). The calculated Systemic Lupus Erythematosus Disease Activity Index (SLEDAI) was 23 (a severe SLE flare). After ruling out malignancy and infections, the patient was diagnosed with MAS in an SLE flare. The patient met the currently used selected criteria for primary and secondary HLH/MAS (Table 2).

As soon as we suspected MAS in the patient suffering from SLE, we started treatment with a high dose of methylprednisolone (MP) (250 mg per day for 3 days) and then continued with 60 mg of oral prednisone per day, reintroduced 400 mg of hydroxychloroquine per day (the patient voluntarily stopped taking hydroxychloroquine and prednisone for the last 6 months), started 100 mg of acetylsalicylic acid daily (previously positive antiphospholipid antibodies) and empirically introduced broad-spectrum antibiotics and acyclovir (while waiting for serology and microbiological cultures) with a good response: a full clinical recovery and laboratory normalization. Our patient was referred to an ophthalmologist for a regular checkup because he had previously received long-term therapy with antimalarials. The best-corrected visual acuity was 20/20 in both eyes with normal intraocular pressures and an unremarkable anterior segment examination. A fundus examination revealed intraretinal hemorrhages, which were more notable in the temporal part of the right eye without vitritis or any exudation in the macular area (Figure 1).

In the right eye, fluorescein angiography revealed diffuse arterial occlusion with extensive capillary non-perfusion within temporal parts of the fundus, while the left eye examination was unremarkable. Due to the MAS and retinal vascular changes without sight disturbances and a good response to the above-mentioned treatment, we opted for concomitant therapy with mycophenolate mofetil (MMF) 2 g per day (he was gastrointestinal intolerant to a higher dose) after discharge. After 4 months of therapy, when the prednisone dose was 7.5 mg per day, the patient noticed visual disturbances, but had good visual acuity without inflammatory changes on the anterior part of the eye. A fundus examination in the right eye revealed temporal intraretinal hemorrhages and vascular sheathing, and in the nasal part of the left eye we observed intraretinal hemorrhages, vascular sheathing and a cluster of neovascularization. A fluorescein angiography of both eyes revealed areas of retinal non-perfusion, irregular retinal artery caliber, arterial and venous dye leakage and the existence of neovascularization in the left eye (Figure 2 and Figure 3).

Leukopenia (3.8 × 10^9^/L), normocytic anemia (hemoglobin 132 g/L), thrombocytopenia (104 × 10^9^/L), and an increase in d-dimers (2300 μg/L), ferritin (548 ng/mL), trygliceride (1.8 mmol/L), AST (54 U/L) and hypofibrinogenemia (2.1 g/L) also recurred. The patient had increased levels of anti-dsDNA, very low complement levels and arthralgias with fatigue. The calculated SLEDAI at that moment was 16 (a severe SLE flare) and the HScore was 135 points (Scores >169 are 93% sensitive and 86% specific for HLH). We assumed that if his prednisone dose was reduced further, he might again experience MAS in addition to a severe SLE flare. We started eye treatment with intravitreal application of BEV in the left eye due to neovascularization and continued with laser photocoagulation in areas of non-perfusion in both eyes, in parallel with RTX 1 g on days 0 and 14, and discontinued MMF with further slow tapering followed by discontinuation of prednisone. Now, 12 months later, our patient is in remission and still has good visual acuity of 20/20 without any visual sensations. On the retina of both eyes, laser-treated retinal zones are clearly visible, without signs of vasculitis, retinal hemorrhages or inflammatory exudates (Figure 4). We decided to maintain remission with RTX 500 mg every 6 months for a minimum of two years due to the patient’s youth and previously inadequate response to MMF.

## 3. Discussion

Here we describe, to our knowledge, the first case of retinal vasculopathy and MAS parallel in SLE flare, treated concomitantly with intravitreal anti-VEGF, laser photocoag-ulation and RTX with a good response. Establishing the diagnosis of MAS in SLE is challenging due to overlapping clinical features (splenomegaly, hepatomegaly, and lymphadenopathy) and laboratory findings (cytopenias) of SLE and MAS. The fact that both conditions mimic each other can lead to a delay in diagnosis and treatment. Hemophagocytosis was not found in the bone marrow aspirate of our patient, but we made the diagnosis of MAS in SLE-flare because our patient fulfilled the currently available criteria for both primary HLH and secondary HLH/MAS [9,10,11,12]. The criteria for MAS in the adult SLE population are not yet developed, so other criteria could be used for the establishment of the diagnosis [13]. Other MAS-specific laboratory parameters found were elevated ferritin, triglyceride, d-dimers, AST, ALT, and LDH, as well as decreased albumin and fibrinogen, and these parameters, particularly hyperferritinemia, should lead rheumatologists to confirm or rule out MAS [6,11,13,14]. A delayed diagnosis of MAS due to the absence of hemophagocytosis in the bone marrow aspirate can lead to a fatal outcome, so it is necessary to start treatment even though only a suspicion of MAS was made. Macrophage activation syndrome is considered a rheumatological emergency and a lethal complication of SLE. Recent studies showed that MAS in the adult SLE population carried a better prognosis than other secondary HLH, with mortality between 5% and 35% [7,15,16]. High mortality is a consequence of insufficiently early recognition of the clinical syndrome, the severity of the disease itself, and the lack of treatment recommendations [17]. Because of the rare occurrence of MAS in the adult SLE population and other rheumatological diseases, management, and treatment recommendations for MAS in adults with rheumatological conditions are not yet developed. Treatment of MAS is so far based on a rheumatologist’s individual assessment of the patient’s illness, previously reported case reports and expert opinions, and the hematological HLH-2004 protocol with dexamethasone, cyclosporine, and etoposide, which can be considered in selected cases [13,17]. Based on the treatment of our patient, but also on previously reported case reports and small clinical studies, our opinion is that the treatment of MAS in SLE should be individually tailored to each patient depending on which target organ (kidney, brain, lungs, eyes, etc.) is affected in parallel by SLE and how severe and life-threatening this comorbidity is. High-dose glucocorticoids present fundamental therapy, and other immunosuppressants, such as MMF, cyclophosphamide, RTX, anti-TNF-alfa, anakinra, canakinumab, immunoglobulins, and plasma exchange are added in cases of severe organ-threatening or refractory disease [13,14,16]. In the beginning, our patient had retinopathy with preserved visual acuity in parallel with MAS. Because of previously published cases and small studies of patients with SLE and MAS and patients with SLE and RV, we first opted for high-dose corticosteroid therapy, which resulted in immediate clinical and laboratory response, and concurrent MMF as an immunosuppressant and glucocorticoid-sparing drug [14,16,18,19,20]. Despite this therapy, the patient had an SLE flare with the possibility of MAS recurrence, significant worsening of RV, and a threatening exacerbation which tends to affect the macula. The treatment of lupus retinopathy depends on the severity of the disease, and the visual outcome was usually better in those with cotton-wool spots than in severe retinal vaso-occlusive disease [21]. Bevacizumab should be considered in severe vaso-occlusive retinopathy, as in our patient. In addition, vitrectomy and retinal photocoagulation can be performed in selected cases to halt neovascularization and prevent aggravation of visual loss [22]. Intraocular therapy in our patient with BEV and laser photocoagulation halted the further progression of retinal damage and preserved visual acuity. Due to SLE flare and RV despite MMF, we decided on a second immunosuppressant RTX in parallel with BEV and laser photocoagulation. We opted for RTX because it has been successfully used for the treatment of MAS and/or RV within SLE [14,15,23,24,25,26,27]. Cyclophosphamide could also be the therapy of choice, but our patient refused it due to unacceptable side effects. After 12 months of RTX therapy, our patient doesn’t have active retinal disease or lupus. According to some authors, retinal vasculitis is a sign of severe SLE disease activity [7,28]. Our case suggests that all patients with severe SLE disease activity should be carefully evaluated for ocular manifestations and clinical features and laboratory parameters of MAS.

## 4. Conclusions

In patients with life-threatening diseases, such as MAS in an SLE flare, and RV, the choice of treatment could be MP with proven fast immunosuppression and RTX with a sustained response given its long therapeutic effect, in combination with intravitreal anti-VEGF and photocoagulation therapy. The goal of the therapy is successful systemic immunosuppression, and intravitreal anti-VEGF therapy in combination with laser photocoagulation is needed in severe cases of RV that demand rapid action until systemic immunosuppressive therapy achieves its full effect.

## Figures and Tables

**Figure 1 ijms-24-02594-f001:**
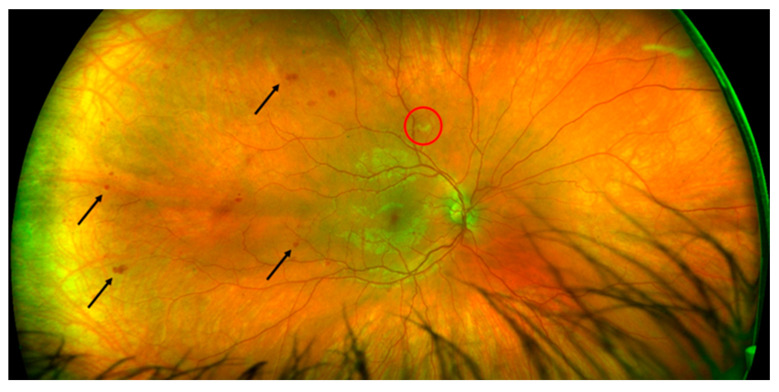
Fundoscopic examination of the right eye showing retinal vasculopathy with hemorrhages (arrows) and cotton wool exudate (red circle).

**Figure 2 ijms-24-02594-f002:**
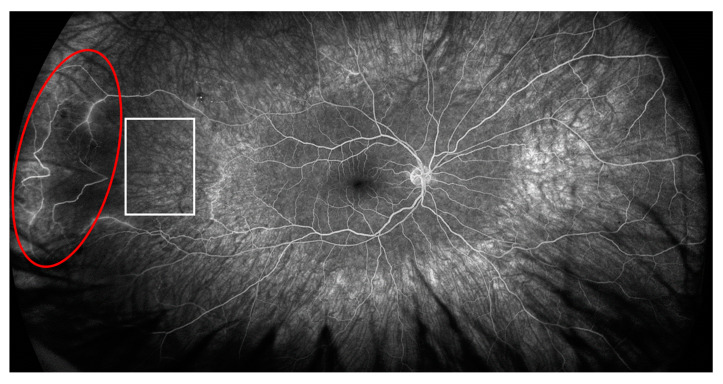
Fluorescein angiography of the right eye showing ischemic area (white square) and area of active vasculitis (red circle).

**Figure 3 ijms-24-02594-f003:**
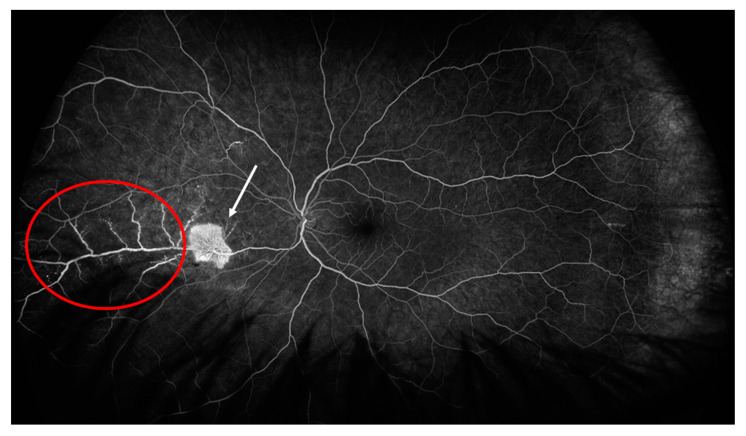
Fluorescein angiography of the left eye showing area of active vasculitis (red circle) and neovascularization (white arrow).

**Figure 4 ijms-24-02594-f004:**
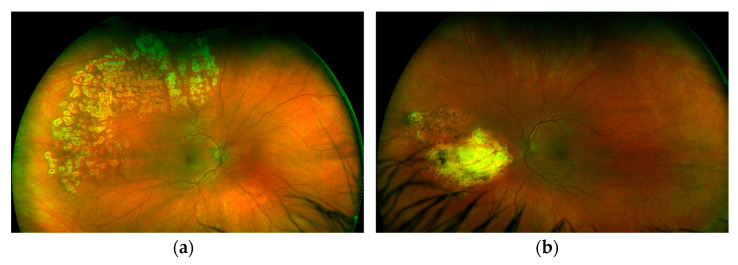
Fundoscopic examination of left (**a**) and right (**b**) eye showing areas of laser scarring.

**Table 1 ijms-24-02594-t001:** Laboratory data upon admission and 12 months after initiation of rituximab (RTX) therapy.

	Reference Range	Admission	After 12 Months of RTX
White blood cell count (10^9^/L)	3.4–9.7	0.79	4.2
Hemoglobin (g/L)	138–175	126	147
Platelets (10^9^/L)	158–424	95	181
ALT (U/L)	12–48	151	32
AST (U/L)	11–38	232	56
LDH (U/L)	<241	666	238
Triglyceride (mmol/L)	<1.7	3.58	0.9
Albumin (g/L)	41–51	32	47
Ferritin (ng/mL)	22–322	4835	130
Fibrinogen (g/L)	1.8–3.5	2.2	3.6
d-dimer (μg/L)	<500	>4530	2960
CRP (mg/dL)	<5.0	25.9	5.9
ESR (mm/3.6 ks)	2–13	10	2
Procalcitonin (µg/L)	0.5–2	0.206	-
Creatinine (µmol/L)	64–104	65	88
24-h urine protein (mg/24 h)	<150	308	172
Direct antiglobulin test	-	Positive	-
C3 (g/L)	0.89–1.87	0.366	0.6
C4 (g/L)	0.17–0.38	0.027	0.05
Antinuclear antibody *	Positive > 1	20	-
Anti-dsDNA antibody * (IU/mL)	Positive > 15	106	175
Anticardiolipin antibody IgM * (MPL-U/mL)	Positive > 40	12	-
Anticardiolipin antibody IgG * (GPL-U/mL)	Positive > 40	3.3	-
Beta2-Glycoprotein I IgM * (EliA U/mL)	Positive > 10	19	-
Beta2-Glycoprotein I IgG * (EliA U/mL)	Positive > 10	3.3	-
Lupus anticoagulant	-	Positive	-

ALT Alanine transaminase, AST Aspartate transaminase, LDH Lactate dehydrogenase, CRP C-reactive protein, ESR Erythrocyte sedimentation rate, * FEIA Phadia 200.

**Table 2 ijms-24-02594-t002:** Selected criteria for MAS/HLH currently in use and applied to our patient.

	Primary HLH	Secondary HLH and MAS		
	HLH-2004, Henter et al. [9]	HScore, Fardet et al. [10]	PRINTO criteria, Ravelli et al. [11]	MAS complicating juvenile SLE, Parodi A et al. [12]
Target population	Primary HLH	Adults	sJIA	Juvenile SLE
Clinical features				
Fever *	+	<38.4 (0), 38.4–39.4 (33), >39.4 (49)	+	+
Hepatomegaly *		Neither (0), either hepatomegaly or splenomegaly (23), both (38) +		+
Splenomegaly *	+			+
Immunosuppression		No (0), yes (18)		
Hemorrhagic manifestations				+
Central nervous system dysfunction				+
Laboratory criteria				
Cytopaenia in more than two lineages *	Either: haemoglobin < 90 g/L, platelets < 100 × 10^9^/L, neutrophils < 1 × 10^9^/L	One lineage (0), two lineages (24), three lineages (34)		White blood cell count ≤ 4.0 × 10^9^/L, hemoglobin ≤ 90 g/L, or platelet ≤ 150 × 10^9^/L
Platelets *			≤181 × 10^9^/L	
Ferritin, ng/mL *	≥500	<2000 (0), 2000–6000 (35), >6000 (50)	>684	>500
Hypertriglyceridaemia, mmol/L *	≥3	<1.5 (0), 1.5–4 (44), >4 (64)	>1.76	2.01
Hypofibrinogenaemia, g/L *	≤1.5	>2.5 (0), <2.5 (30)	≤3.6	≤1.5
Liver function tests, IU/L *		AST < 30 (0), >30 (19)	AST > 48	AST > 48, LDH > 567
Low/absent NK cell activity	+			
Soluble CD25, U/ml	≥2400			
Haemophagocytosis	+	No (0), yes (35)	+	+
Fulfillment of criteria	Molecular diagnosis consistent with primary HLH or five or more of eight criteria	Produces a probability outcome. Scores > 169 are 93% sensitive and 86% specific for HLH	Febrile patient with known or suspected sJIA, ferritin > 684 ng/mL and two or more additional items	Diagnosis of MAS if one clinical + two laboratory criteria or evidence of hemophagocytosis in the bone marrow aspirate
Our patient	Meets the criteria	HScore 223 points (96–98% probability of HLH)	Meets the criteria	Meets the criteria

PRINTO Paediatric Rheumatology International Trials Organization, HLH haemophagocytic lymphohistiocystosis, SLE systemic lupus erythematosus, AST aspartate transaminase, LDH lactate dehydrogenase, sJIA systemic-onset JIA, MAS macrophage activation syndrome, * clinical features and laboratory parameters present in our patient upon admission, + clinical features and laboratory findings included in the selected HLH/MAS criteria.

## Data Availability

The original data generated and analyzed for this study are included in the published article. Further inquiries can be directed to the corresponding author.

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
