# Peer review of "Rituximab, Intravitreal Bevacizumab and Laser Photocoagulation for Treatment of Macrophage Activation Syndrome and Retinal Vasculitis in Lupus: A Case Report"

_ijms, 2023, doi:10.3390/ijms24032594_

Round 1

Reviewer 1 Report

It si an interesting case of male SLE patient

with rare manifestations - retinal vasculitis

and MAS.

In introduction about case I missed the

sentences about patient medical history (Sle 

start day, therapy,…) prior discribed flare.

I dont have any further major comments.

Reviewer 2 Report

The authors describe the first case of retinal vasculopathy and MAS 138 parallel in SLE flare, treated concomitantly with intravitreal anti-VEGF, laser photocoagulation, and RTX with a good response. MAS is considered a rheumatological emergency 140 and a lethal complication of SLE, thus it is really interesting to see what are treatment options for such a condition.

The case report is clearly structured and presented with interesting pictures.

I strongly recommend accepting the manuscript for publication, and also strongly encourage authors for the next step i.e. randomized study in order to compare different treatment options in patients with MAS. 

Reviewer 3 Report

Dear Authors and Reviewers! Thank you so much for the opprtunity to reviewer the manuscript. Herein Authors describe a rare complicated case of SLE with MAS and RV. Both conditions are not so frequent in SLE , but have been described before. The combination of two complcation at one patient in my opinion has not been previously published yet. 

The diagnostics of MAS in SLE is a challenging problem, due to ansence of validated criteria for SLE, compare to HLH and sJIA and the main problem is overlapping lupus and MAS features, so it is very difficult to distinguishe MAS from SLE and this case has the same problem.

The absence of bone marrow aspirate also make the dicision difficult.

I think authors should make focus in the manuscript on MAS identification in their patient, may be provide the table with MAS and SLE features or provide the table where authors check if patien had or now to known MAS crieteria, eg HLH, Hscore, Parodi's criteria for pediatric SLE (young adult), also authors can provide the Hscore in different key time point. The discussion also should be focused on difficulties of distiguishe MAS from SLE.

The introduction has unneccessary sentences about sex differences in SLE.

Round 2

Reviewer 3 Report

The manuscript has became better and might be accepted. Authors responded well to my comments. Thanks a lot!